# The Role of Innovation Capacities in the Relationship between Green Human Resource Management and Competitive Advantage in the Saudi Food Industry: Does Gender of Entrepreneurs Really Matter?

**Hassane Gharbi** [1,2,3], **Abu Elnasr E. Sobaih** [1,2,4,*] , **Nadir Aliane** [1,2] and **Ayth Almubarak** [1,5]

1  The Saudi Investment Bank Scholarly Chair for Investment Awareness Studies, The Deanship of Scientific Research, The Vice Presidency for Graduate Studies and Scientific Research, Al-Ahsaa 31982, Saudi Arabia; hgharbi@kfu.edu.sa (H.G.); nhaliane@kfu.edu.sa (N.A.); aymubarak@kfu.edu.sa (A.A.)
2  Management Department, College of Business Administration, King Faisal University, Al-Ahsaa 31982, Saudi Arabia
3  Management Department, School of Business, University of Sfax, Sfax 3018, Tunisia
4  Hotel Management Department, Faculty of Tourism and Hotel Management, Helwan University, Cairo 12612, Egypt
5  Accounting Department, College of Business Administration, King Faisal University, Al-Ahsaa 31982, Saudi Arabia
*  Correspondence: asobaih@kfu.edu.sa

**Abstract:** Adopting environmentally friendly behavior has become more than a claim. Green human resource management seems to be the solution where innovation will be a strategic lever to lead the company, with green practices, to the possession of a decisive competitive advantage. The purpose of this research is to examine the mediating role of innovation capacities in the relationship between green human resource management and competitive advantage in the Saudi food industry. The research compares between males and females in this relationship. For this purpose, we have used a quantitative approach to conduct the research. Using a sample of 1114 female and male entrepreneurs, owner–managers of small and medium different food companies, operating in the Saudi territory, especially in the major cities, namely Riyadh, Medina, Makkah, Sharaqiyah, Tabuk, Al Qasim and Najran. We were able to make a gender comparison of the mediating role of innovation in the above relationship. The results of the structural equation modelling (SEM) via AMOS software version 23 showed a perfect mediation of the innovation capacities for female entrepreneurs, and partial mediation for male entrepreneurs in the relationship between green human resource management and competitive advantage. Following a focus group with ten female and male entrepreneurs, we were able to understand the reasons for the results we arrived at. The results of our research have numerous implications for both scholars and policymakers, especially in relation to the Saudi food industry.

**Keywords:** green human resource management; innovation capacities; competitive advantage; gender; Saudi food industry; Saudi Arabia

## 1. Introduction

The earth is suffocating is no longer the title of a science fiction film but a bitter reality. Massive deforestation, melting ice in Antarctica, greenhouse gas emissions and global warming are the leitmotif for many researchers around the world who have sounded the alarm on the alarming situation of our planet. Therefore, adopting environmentally friendly behavior [1] has become more than a necessity. In order to rectify the situation and preserve the little hope we have in saving our natural resources, all disciplines have become

involved, including human resource management, which has now become green human resource management (GHRM) [2–6]. Indeed, GHRM can support better implementation of an organization's environmental management [7], while ensuring its success [8]. Companies have realized that the maintenance and even development of their business will now depend on the continuous supply of natural resources [9]. In order to achieve sustainable development, environmental issues have become the main concern of all socially responsible organizations [10] who aim to save our planet. Thus, GHRM practices, in this case green recruitment, training, appraisal, etc., become essential to ensure employees' participation in nature-friendly work activities, while motivating them to be environmentally conscious and use company resources in a sustainable manner [11].

Organizations are increasingly supporting the green innovative practices of their employees to promote sustainable environmental development. Green innovation is a predicator of achieving a competitive advantage for today's companies, allowing them to make more profits over their competitors. The study by Albort-Morant et al. [12] is a clear example of this, which revealed that green innovative organizations are more competitive and globally more successful than their competitors. It was confirmed that environmental disclosure positively and significantly affected the financial performance of the companies, which is also moderated by social and ethical practices [13]. Despite the direct and indirect relationship between GHRM, innovation capacities and competitive advantage being clearly examined in previous research studies (see for example, [4–10]), the role of gender in this relationship was not examined to the best of our knowledge. Understanding this role and the differences between male and female, enables scholars to recognize the heterogeneity of entrepreneurs based on their gender. It will also allow policymakers to have better regulations, whether environmental, social or economic, for their entrepreneurs to achieve the national agenda through collaboration with various stakeholders, which contributes to the development of innovation, meets consumer needs and creates environmental, social and economic value [14].

It is important to note that in our research, a significant percentage of our sample was taken under the aegis of Monsha'at, a governmental body that was established in 2016. Its main objectives are to organize, support, develop and sponsor the small- and medium-sized companies sector with the ultimate goal of increasing their contribution to GDP from 20% to 35% by 2030 (https://www.monshaat.gov.sa/about, accessed on 15 April 2022). For its part, Monsha'at supports programs and projects aimed at disseminating the culture of self-employment and entrepreneurship; initiative and innovation diversifies the sources of financial support for companies.

In this context, the idea of the current research was born to compare the green human resource management practices of female entrepreneurs with that of their male counterparts and, consequently, their respective impacts on the innovation capacities of each gender and on the resulting competitive advantages in the food industry in Saudi Arabia. This will be through entrepreneurial platforms, such as business incubators and co-working spaces. There is a key research question here: is this entrepreneurial culture that Monsha'at is trying to crystallize in the habits of young entrepreneurs assimilated as a process in the same way by female and male entrepreneurs? In other words, do they (male and female) have the same conception of green human resource management? If so, what might be their impact on the role of innovative capacities as a mediating variable linking GHRM to competitive advantage? The answers to these research questions will have various implications for both scholars and policymakers in the Saudi food industry, which are discussed and elaborated. The structure of the current article will be as follows: it will start with discussing the theoretical framework of the research and the research hypothesis. It then discusses the research methodology and presents the research results. It then discusses the results of the research and provides the research implications for both scholars and policymakers. Finally, it presents the research conclusion as well as limitations of the research and opportunities for future research.

## 2. Theoretical Framework and Hypotheses Development

### 2.1. Green Human Resource Management and Innovation

The literature is more interested in the innovative capacity of firms, since it provides them with added value and increases their chances of outperforming the competition [14,15]. Gopalakrishnan and Damanpour [16] define innovation as programs, policies, systems, equipment, services, products, behaviors or ideas newly adapted by the organization [17,18]. Wang, et al. [19] argue that innovation is conceptually a process that starts with a new idea and ends with the introduction to the market [19]. Certainly, many researchers such as Gönül et al. [20] consider innovation as the vehicle for productive change. Thus, there is growing interest in studying the role of GHRM in supporting firms' innovative capacity [21]. However, it remains an activity that needs to be handled with extreme care to reap its benefits as it involves human resources (HR), processes and technology. Indeed, based on the assumption of the Resource-based View Theory (RBV), Barney [22] believes that if organizational resources become depreciated or obsolete, or are imitated, then the competitive position, as well as the rents usually allocated, will disappear. Thus, for RBV, the uniqueness of HR is emphasized and is inherent in their knowledge, abilities, experiences and behaviors within the organization. Indeed, HR remains one of the most important organizational resources that remains difficult to replicate. Chaudhary [23] investigates the role of GHRM in promoting employees' environmental performance. The study shows that GHRM significantly predicts employees' environmental behavior. Thus, based on this assumption, a range of good GHRM practices can advance the creative and innovative behaviors of individuals. Elshaer et al. [24] found that GHRM positively and significantly influences employee environmental performance and overall organizational environmental performance.

As recommended by Chummee [25], organizational innovation depends on employer support. Therefore, top management need to recognize the challenge of developing and implementing an HRM that promotes good work by creating the right context for HR to feel motivated, satisfied and reassured; by being committed to learning and sharing the knowledge, they have gained, along with others, the intention to innovate [26]. In the same vein, Lado and Wilson [27] argue that HR practices can promote and facilitate employee creativity and innovation [27]. More recent research shows that HRM practices have a significant and positive impact on product innovation [28]. Another recent study [29] showed that the GHRM has a positive significant influence on the green innovation of organizations. Thus, our first hypothesis is formed as follows:

**Hypothesis 1 (H1).** *Green human resource management positively affects innovation capacity.*

### 2.2. Green Human Resource Management and Competitive Advantage

According to Malik et al. [30], in addition to regulatory standards, good environmental policies are crucial for building a company's image and competitive advantage. Indeed, human resources are one of the key factors in creating competitive advantage [31,32]. Designated by Tooranloo [33] as the central pillar of an organization's competitive advantage, human resources are recognized as an essential resource that is difficult to imitate by the competition. Competitive advantage is seen as the ultimate goal [34] for any organization that wants to remain in an environment that is increasingly characterized by tough and sometimes unfair competition.

In order to gain and maintain this competitive advantage, companies should have the ability to highlight a greater differential or relative value than its competitors and convey this relevant information to its target audience in the best possible way [35]. In this way, a competitive advantage lies in the company's ability to generate goods/services in a more effective way than its immediate competitors [36]. In this regard, Barney [22] argues that a firm has a competitive advantage when it implements a value-creating strategy that is not simultaneously implemented by any current or potential players. Consequently,

many authors [37–41] believe that human resource management has evolved from a purely bureaucratic administrative position to a strategic function, thereby promoting competitive advantage and value creation for organizations. Thus, our second hypothesis will take the following form:

**Hypothesis 2 (H2).** *Green human resource management positively affects competitive advantage.*

### 2.3. Innovation and Competitive Advantage

Innovation plays a strategic role for a company in building its competitive advantage [42]. Several researchers [43–45] have indicated that innovation has a positive and significant effect on the competitive advantage of the organization. These previous research studies confirmed that sustainable competitive advantage, which every organization wants to have, can be derived from its ability to innovate. From this context, the results found by Mulyono and Suprapto [46] attest that organizational innovation positively and significantly impacts on the competitive advantage of organizations. Furthermore, and in the same vein of ideas, the empirical results of a research undertaken by Dang and Wang [47] show that innovation has a positive influence on the competitive advantage of hotel companies. Innovation was found to have a direct, positive significant influence on a firm's green performance and a mediating role between GHRM and environmental performance [28]. In research carried out by Camison and Puig-Denia [48], it was proved that the emphasis on innovation revealed a positive relationship with the reduction in product failure rate, the reduction in process management complexity and the reduction in complaints made by customers. Thus, based on the above, our third and fourth hypotheses will take the following form:

**Hypothesis 3 (H3).** *Innovation capacities positively affect competitive advantage.*

**Hypothesis 4 (H4).** *Innovation capacities play a mediating role in the relationship between green human resource management and competitive advantage.*

## 3. Research Methodology
### 3.1. Research Population and Sample

A quantitative approach was used to conduct this research. A questionnaire was distributed and collected via the research team's personal network to a sample of 1500 entrepreneurs in the food industry, including food production, processing, service and marketing. Owner–managers of small- and medium-sized food companies, of all genders, operating in Saudi Arabia, especially in the major cities of Riyadh, Medina, Makkah, Sharaqiyah, Tabuk, Al Qasim and Najran, were contacted. We were only able to receive 1114 usable questionnaires, i.e., a 76% return, of which 50.1% were men and 49.9% were women, all of whom were owner–managers of small- and medium-sized companies. We were keen that our collected data should represent both male and female participants to be able to have a fair comparison. In relation to the demographics of our respondents, 65.52% of our sample were graduates, of which 54.79% were women and 45.21% were men. The age range of our sample varies; however, the majority of our participants were aged between 25 and 45 years.

### 3.2. Measurement

Following a review of the literature, measurement scales were selected to form the research questionnaire survey (please see items in Table 1) to enable us to measure the expected variables. The questionnaire items, addressed to the full sample, have minimum and maximum values that range from 1 to 5. The means for all responses range from 2.73

to 4.62, along with standard deviation values that range from 0.423 to 1.901 (see Table 1), indicating that our data are more dispersed and less condensed around the mean value [49].

**Table 1.** The research instrument.

| Scale Items | Variable | Authors |
|---|---|---|
| IN1—Our company frequently tries new ideas. | | |
| IN2—Our company seeks new ways of doing things. | | |
| IN3—Our company is creative in its methods of operation. | Innovation capacity | Calantone et al. [50] |
| IN4—Our company is often the first to market new products and services. | | |
| IN5—Our Company's Innovation meets resistance. | | |
| IN6—Our introduction of new products has increased over the past two years. | | |
| CA7—Our prices per product/service unit are lower than our competitors' prices. | | |
| CA8—We will continuously improve our cost–efficiency. | | |
| CA9—We are cost efficient. | competitive advantage | Cater et al. [51] |
| CA10—Compared with our competitors' products/services, our quality is better. | | |
| CA11—Compared to our competitors, we are faster in meeting the needs of our customers. | | |
| CA12—Compared to our competitors, we are more flexible in meeting the needs of our customers. | | |
| M13—My enterprise provides adequate training to promote environmental management as a core organizational value. | | |
| M14—My enterprise considers how well employee is doing at being eco-friendly as part of their performance appraisals. | | |
| M15—My enterprise relates to employee's eco-friendly behavior to rewards and compensation. | green human resources management | Shen and Benson [52] |
| M16—My enterprise considers personal identity–environmental management fit in recruitment and selection. | | |
| M17—Employees fully understand the extent of corporate environmental policy | | |
| M18—My enterprise encourages employees to provide suggestions on environmental improvement. | | |

We would like to point out that a very specific approach was taken in the first version of the questionnaire. Indeed, we chose measurement scales with as few items as possible to motivate potential respondents to perform their tasks and complete the questionnaire effectively. In addition, some minor improvements to the structure of some items were undertaken after a pilot study with 15 participants. All variables were measured via 5-point Likert-type scales ranging from 1 "strongly disagree" to 5 "strongly agree".

*3.3. Principal Component Analysis and Purification of Measurement Scales*

We started with a principal component analysis (PCA) with a varimax rotation using SPSS software (version 23) to test the quality of the representation. The unidimensionality of the variables "innovation capacities, competitive advantage and green human resource management" is confirmed with the identification of a single component representing, respectively, 57.996% for female entrepreneurs (FE) and 58.137% for male entrepreneurs (ME); 58.326% for FE and 53.410% for ME; and 53.293% for FE and 52.656% for ME of the total variance explained. The KMO index shows values above 0.5, a tolerable threshold according to [53], and the Bartlett tests were significant. Following this, we can conclude that the PCA results showed that our variables were well suited for factoring. For reliability, Cronbach's Alpha was used. The results showed that the Alpha values were almost excellent [54]. Moreover, the *p*-Value for the three variables is equal to zero, hence the rejection of the null hypothesis. Finally, the PCA allowed us to eliminate certain items that were weakly related to their basic construct, in this case, items IN1-IN2-IN4-IN5-M17-M18-M19 for the FE and items IN4-IN5-CA8-CA9-CA11-M14-M19 for the ME.

## 4. Results

*4.1. Confirmatory Factor Analysis*

Confirmatory factor analysis (CFA) was used to test the fit of the selected scale to the data collected. The criteria for interpreting the results of a confirmatory factor analysis are numerous, often grouped into three categories. Firstly, the absolute fit indexes, which make it possible to evaluate the extent to which the theoretical model correctly reproduces the data collected. This is notably the case for the $x^2/ddl$ parsimony index, whose value must be less than 5 [55], the SRMR, whose value must be less than 0.05, and the RMSEA, whose value must be less than 0.08 and, if possible, 0.05 [56]. Then, the incremental indexes, which are used to assess the improvement in the fit of the model, being tested in comparison to a more restrictive reference model. More specifically, these are the NFI, the TLI and the CFI with a threshold value of 0.90 [57]. Finally, there are the parsimony indexes via the normalized $X^2$.

The results of the first-order confirmatory factor analysis combining the dependent and independent variables of the study fit the data (Table 2). They show a Khi$^2$ to its degree of freedom $x^2/ddl$ (2.700) for the FE and an $x^2/ddl$ equal to (1.533) for the ME. These ratios are considered satisfactory since they are lower than three. Moreover, the RMSEA indexes for the FE and ME have respective values of 0.055 and 0.031, thus approaching zero, showing that the quality of the adjustments is appropriate. The indexes NFIf = 0.938 and NFIm = 0.937, TLIf = 0.939, TLIm = 0.969, CFIf = 0.959 and CFIm = 0.977 also testify to the values admitted by the literature to offer a very good fit to our first order models. The results of the exploratory factor analysis of the latter thus meet the recommended standards [58].

Two indicators are provided in the literature, namely the skewness coefficient and the kurtosis coefficient, to compare the observed distribution with the normal distribution or Gauss curve. The skewness coefficient "shows whether the observations are distributed equitably around the mean (the coefficient is then zero) or whether they are rather concentrated towards the lowest values (positive coefficient) or whether they are rather concentrated towards the highest values (negative coefficient)" [59]. The Kurtosis coefficient compares "the shape of the distribution curve of the observations to that of the normal distribution: a positive coefficient indicates a higher concentration of observations, while a negative coefficient indicates a flatter curve" [59]. In our case, the symmetry (Skewness) and kurtosis coefficients do not violate the normality assumption [60] and show admissible values. We can conclude in this respect that all distributions are fairly distributed and all variables follow the normal distribution (Table 2).

In order to know whether the items of our constructs, which are supposed to measure the same phenomenon, are correlated, we had to calculate their convergent validities. This was done through the Composite Reliability (CR), which must be strictly greater than 0.7, and the Average Variance Extracted (AVE), which must be strictly greater than 0.5. The results (see Table 3) show that convergent validity was verified for all variables [61] for both the FE and their male counterparts. To find out whether two theoretically distinct variables are also distinct in practice, we calculated the discriminant validity. This involved checking whether the square root of the AVE of each variable is strictly greater than the correlations it shares with the other variables. The results (Table 3) confirm that the discriminant validity was verified for all three variables, namely "innovation capacity, competitive advantage and human resources management", for both FE and ME.

**Table 2.** Descriptive statistics for female and male entrepreneurs.

| Abbr | Item | Min | Max | M | SD | Skewness | Kurtosis |
|---|---|---|---|---|---|---|---|
| | Innovation Capacity | | | | | | |
| IN4 | Our company is often the first to market new products and services. | 1 | 5 | 3.98 | 0.881 | −1.009 | 1.385 |
| IN6 | Our introduction of new products has increased over the past two years. | 1 | 5 | 2.73 | 1.000 | 0.035 | 3.90 |
| | Comp. Advantage | | | | | | |
| CA7 | Our prices per product/service unit are lower than our competitors' prices. | 1 | 5 | 3.66 | 1.437 | −0.541 | −0.231 |
| CA8 | We will continuously improve our cost–efficiency. | 1 | 5 | 3.80 | 0.909 | −0.801 | −0.069 |
| CA9 | We are cost efficient. | 1 | 5 | 4.09 | 0.910 | −1.104 | −1.610 |
| CA10 | Compared with our competitors' products/services, our quality is better. | 1 | 5 | 4.06 | 0.866 | −1.018 | −1.192 |
| CA11 | Compared to our competitors, we are faster in meeting the needs of our customers. | 1 | 5 | 4.23 | 0.888 | −1.192 | 1.063 |
| CA12 | Compared to our competitors, we are more flexible in meeting the needs of our customers. | 1 | 5 | 4.01 | 0.804 | −0.973 | 0.891 |
| | HRM.946 | | | | | | |
| M13 | My enterprise provides adequate training to promote environmental management as a core organizational value. | 1 | 5 | 4.06 | 0.856 | −1.055 | 1.660 |
| M14 | My enterprise considers how well employee is doing at being eco-friendly as part of their performance appraisals. | 1 | 5 | 3.98 | 0.849 | −0.829 | 0.869 |
| M15 | My enterprise relates to employee's eco-friendly behavior to rewards and compensation. | 1 | 5 | 3.58 | 0.950 | −0.278 | −0.202 |
| M16 | My enterprise considers personal identity–environmental management fit in recruitment and selection. | 1 | 5 | 4.32 | 1.010 | −1.595 | 1.001 |
| | Innovation Capacity | | | | | | |
| IN1 | Our company frequently tries new ideas. | 1 | 5 | 3.83 | 1.018 | −0.868 | 0.373 |
| IN2 | Our company seeks new ways of doing things. | 1 | 5 | 3.79 | 0.967 | 0.763 | 0.198 |
| IN3 | Our company is creative in its methods of operation. | 1 | 5 | 3.98 | 0.881 | −1.009 | −1.385 |
| IN6 | Our introduction of new products has increased over the past two years. | 1 | 5 | 3.73 | 1.901 | −0.816 | −0.390 |
| | Comp. Advantage | | | | | | |
| CA7 | Our prices per product/service unit are lower than our competitors' prices. | 1 | 5 | 3.51 | 0.909 | −0.640 | −0.951 |
| CA10 | Compared with our competitors' products/services, our quality is better. | 1 | 5 | 4.06 | 0.888 | −1.018 | 1.192 |
| CA12 | Compared to our competitors, we are more flexible in meeting the needs of our customers. | 1 | 5 | 4.01 | 0.946 | −0.973 | −0.891 |
| | GHRM | | | | | | |
| M13 | My enterprise provides adequate training to promote environmental management as a core organizational value. | 1 | 5 | 3.48 | 1.234 | −0.037 | −1.482 |
| M15 | My enterprise relates to employee's eco-friendly behavior to rewards and compensation. | 1 | 5 | 3.46 | 1.532 | 0.025 | −1.654 |
| M16 | My enterprise considers personal identity–environmental management fit in recruitment and selection. | 1 | 5 | 4.05 | 0.664 | −0.460 | −1.013 |
| M17 | My enterprise considers personal identity–environmental management fit in recruitment and selection. | 1 | 5 | 4.62 | 0.423 | −0.324 | 0.823 |
| M18 | Employees fully understand the extent of corporate environmental policy. | 1 | 5 | 3.78 | 1.821 | −1.201 | −0.605 |

Female Model fit: ($\chi^2$ (44, N = 556) = 118.822 $p < 0.001$, normed $\chi^2$ = 2.700, RMSEA = 0.055, SRMR = 0.0361, CFI = 0.959, TLI = 0.939, RFI = 0.906, IFI = 0.960, NFI = 0.938, PCFI = 0.640 and PNFI = 0.625). Male Model fit: ($\chi^2$ (49, N = 558) = 75,105 $p < 0.001$, normed $\chi^2$ = 1.533, RMSEA = 0.031, SRMR = 0.0349, CFI = 0.977, TLI = 0.969, RFI = 0.916, IFI = 0.977, NFI = 0.937, PCFI = 0.725 and PNFI = 0.696).

**Table 3.** Convergent and discriminant validity (developed by authors).

| Factors and Items | Standardized Loading | | CR | | AVE | | MSV | | 1 | | 2 | | 3 | |
|---|---|---|---|---|---|---|---|---|---|---|---|---|---|---|
| | F | M | F | M | F | M | F | M | F | M | F | M | F | M |
| 1-Innovation Capacity (α = 0.877) | | | 0.905 | 0.959 | 0.826 | 0.854 | 0.421 | 0.446 | **0.908 *** | **0.924** | | | | |
| IN1 | | 0.879 | | | | | | | | | | | | |
| IN2 | | 0.914 | | | | | | | | | | | | |
| IN3 | 0.908 | 0.974 | | | | | | | | | | | | |
| IN4 | | | | | | | | | | | | | | |
| IN5 | | | | | | | | | | | | | | |
| IN6 | 0.910 | 0.928 | | | | | | | | | | | | |
| 2-Comp. Advantage (α = 0.962) | | | 0.948 | 0.934 | 0.754 | 0.825 | 0.624 | 0.677 | 0.766 | 0.771 | **0.868** | **0.908** | | |
| CA7 | 0.966 | 0.907 | | | | | | | | | | | | |
| CA8 | 0.982 | | | | | | | | | | | | | |
| CA9 | 0.831 | | | | | | | | | | | | | |
| CA10 | 0.876 | 0.950 | | | | | | | | | | | | |
| CA11 | 0.777 | | | | | | | | | | | | | |
| CA12 | 0.753 | 0.866 | | | | | | | | | | | | |
| 3-GHRM (α = 0.938) | | | 0.908 | 0.957 | 0.764 | 0.818 | 0.624 | 0.677 | 0.556 | 0.565 | 0.771 | 0.801 | **0.874** | **0.904** |
| M13 | 0.803 | 0.826 | | | | | | | | | | | | |
| M14 | 0.853 | | | | | | | | | | | | | |
| M15 | 0.930 | 0.890 | | | | | | | | | | | | |
| M16 | 0.904 | 0.910 | | | | | | | | | | | | |
| M17 | | 0.953 | | | | | | | | | | | | |
| M18 | | 0.839 | | | | | | | | | | | | |
| M19 | | | | | | | | | | | | | | |

* Please note bold diagonal values: the square root of AVE for each dimension; below diagonal values: inter-correlation between dimensions.

To calculate discriminant validity, we will need the correlation matrix, the square roots of the AVEs and the α Cronbach's for each variable (Table 3). The square roots of the AVEs are greater than the off-diagonal values, which represent the correlations between these variables; thus, confirming the discriminant validity of the factors as presented by Fornell and Larcker [62]. Furthermore, the mean extracted variance (AVE) scores, specific to the female entrepreneurship model, for innovativeness (0.826), competitive advantage (0.754) and human resource management (0.764), are well ahead of the maximum shared variances (MSV), which represent the following values (0.421, 0.624, 0.624), respectively. Similarly, for the male entrepreneurs' model where the AVE scores for innovative ability (0.854), competitive advantage (0.825) and human resource management (0.818) clearly outperform the maximum shared variances (MSV), which represent the following values (0.446, 0.677, 0.677), respectively. Subsequently, as suggested by Hair et al. [63] discriminant validity is ensured for both models. Furthermore, the inter-correlation scores for each variable must not be greater than the values on the diagonal indicating the square roots of AVEs for each specific factor. A detail that has been honored (see Table 3, in bold).

### 4.2. Results of Structural Equation Modelling

As soon as the validity and reliability of the measures are calculated and verified, we can move on to structural equation modelling to test the impact of green human resource management on competitive advantage for both female and male entrepreneurs via the innovation capacities of each category.

First, the results from the female entrepreneurs' model fit the data (Table 4). They show a khi$^2$ related to its degree of freedom x$^2$/ddl equal to (2.768). This ratio is considered satisfactory since it is less than 3. Moreover, the RMSEA index is equal to 0.056; as it is approaching zero, it shows us that the fit is satisfactory. The indexes NFI = 0.935, TLI = 0.936 and CFI = 0.957 also authenticate the values admitted by the literature to offer a very good fit. The square root of the adjusted residuals average RMR = 0.029 and the standardized RMR, SRMR = 0.035 are found to be excellent as they are very close to zero. All of the above hypotheses were tested and show significant relationships with $p < 0.001$ and $p < 0.05$ (Table 4, Figure 1). Specifically, FE green human resource management has a significant and positive effect on innovation capacities (β = +0.62, $p < 0.001$) and a significant and positive effect on competitive advantage (β = +0.81, $p < 0.001$). As for innovation capacity, it has a

significant and positive effect on competitive advantage ($\beta$ = +0.17, $p$ < 0.05). Moreover, the robustness of the structural model is further legitimized by the significant coefficient of the value of ($R^2$ = 0.753) (see Table 4), which in our case represents the proportion of competitive advantage explained by human resources management and innovation capacity in the regression model. Indeed, by using human resources management and innovation capacity, we can explain about 76% of the variance of the competitive advantage.

**Table 4.** Result of the female structural model (developed by authors).

| Result of the Structural Model | $\beta$ | C-R T-Value | $R^2$ | Hyp. Results |
|---|---|---|---|---|
| H1—FGHRM $\rightarrow$ INNOVACAP | 0.62 *** | 26.161 | | Supported |
| H2—INNOVACAP $\rightarrow$ COMPAD | 0.17 ** | 23.949 | | Supported |
| H3—FGHRM $\rightarrow$ COMPAD | 0.81 *** | 13.575 | | Supported |
| COMPAD | | | 0.753 | |

Model fit: ($\chi^2$ (45, N = 556) = 124,542 $p$ < 0.001, normed $\chi^2$ = 2768, RMSEA = 0.056, RMR = 0.029, SRMR = 0.0352, GFI = 0.964, CFI = 0.957, NFI = 0.935, RFI = 0.904, TLI = 0.936, PCFI = 0.652 and PNFI = 0.637), *** $p$ < 0.001; ** $p$ < 0.01.

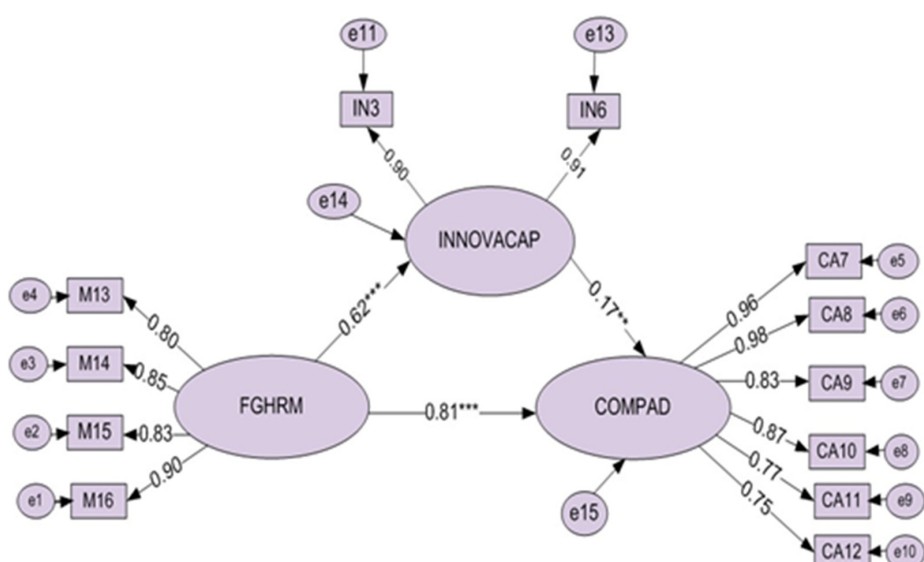

**Figure 1.** Structural model for females (perfect mediation of innovation capacity). *** $p$ < 0.001; ** $p$ < 0.01.

Secondly, it should also be noted that the results from the male entrepreneurs' model fit the data (Table 5). They reveal a Khi$^2$ related to its degree of freedom $x^2$/ddl equal to (1.774). This ratio is considered satisfactory since it is less than three. Moreover, the RMSEA index is equal to 0.037; as it is approaching zero, it shows us that the fit is satisfactory. The indices NFI = 0.926, TLI = 0.955 and CFI = 0.966 also authenticate the values admitted by the literature to offer a very good fit. The square root of adjusted residuals average RMR = 0.022 and the standardized RMR, SRMR = 0.0376 are found to be excellent as they are very close to zero. All of the above hypotheses were tested and show significant relationships with $p$ < 0.001 and $p$ < 0.05 (see Table 5 and Figure 2). More concretely, green human resource management specific to male entrepreneurs has a significant and positive effect on their innovation capacities ($\beta$ = +0.20, $p$ < 0.05) and a significant and positive effect on competitive advantage ($\beta$ = +0.13, $p$ < 0.05). Regarding innovation capacity, it has a significant and positive effect on competitive advantage ($\beta$ = +0.70, $p$ < 0.001). Moreover, the robustness of the structural model is further justified by the significant coefficient of the value of ($R^2$ = 0.723) (Table 5), which in our case represents the proportion of competitive advantage explained by the green management of human resources and innovation capacity in the regression model. Indeed, using green human resource management and innovative capacity, we can explain about 73% of the variance in competitive advantage.

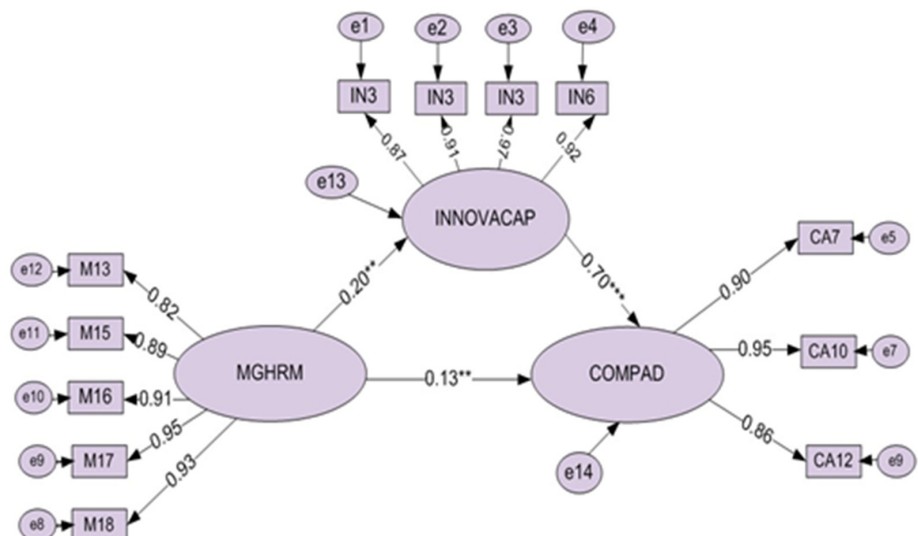

**Figure 2.** Structural model for males (partial mediation of innovation capacity). *** $p < 0.001$; ** $p < 0.01$.

The approach of Baron and Kenny [64] has been of great help to us in order to verify and approve the mediating role of innovative capacity in the relationship between green human resource management and competitive advantage. This approach consists of a series of four consecutive tests that we will test on the two models mentioned above. First, we need to demonstrate that the link between green human resource management and competitive advantage is significant to ensure that there is a potential impact to be mediated. Indeed, the models for female and male entrepreneurs, respectively, show that green human resource management has a significant and positive effect on competitive advantage ($\beta = +0.81$, $p < 0.001$), ($\beta = +0.13$, $p < 0.05$). Moreover, respectively, in the regression of the competitive advantage on green human resource management, the coefficients are significant (with Student's test values equal to $3.625 \geq 1.96$; $p = 0.05$ and $2.703 \geq 1.96$).

For both models, we need to demonstrate that green human resource management has a significant impact on the mediator variable; here in our case is innovation capacity, considered as an exogenous variable in a regression analysis of the innovation capacity on green human resource management. Indeed, both models show that green human resources management has a significant and positive effect on innovation capacity ($\beta = +0.62$, $p < 0.001$) and ($\beta = +0.20$, $p < 0.05$) (see Tables 4 and 5).

Third, we need to show that the link between the mediator variable or innovative capacity and competitive advantage is significant for both models. The evidence shows that innovative capacity has a significant and positive effect on competitive advantage ($\beta = +0.17$, $p < 0.05$) and ($\beta = +0.70$, $p < 0.001$). In addition, the competitive advantage is regressed on both innovation capacity and green human resource management. By controlling for the latter, the coefficient between innovation capacity and competitive advantage must remain significant for both models (see Table 3).

Finally, we are led to verify the partial or perfect nature of the capacity for innovation by examining the significance of the direct links between the green management of human resources and the competitive advantage (see Tables 6 and 7). Indeed, using the boostrapping technique, which the Amos software (version 23) offers us, in this case "The user-defined estimands"; it is important to mention that Table 6 reveals a piece of crucial information. It shows that the link between green human resource management and competitive advantage is no longer significant after the introduction of the mediator variable ($\beta = +0.113$, $p = 0.056 > 0.05$), whereas it was significant in the first stage of Kenny and Baron's approach ($\beta = +0.81$, $p < 0.001$). Therefore, we find that the mediation by the innovative capacity proposed by female entrepreneurs is thus complete between green human resource management and competitive advantage.

**Table 5.** Result of the male structural model (developed by authors).

| Result of the Structural Model | β | C-R T-Value | R² | Hyp. Results |
|---|---|---|---|---|
| H1—MGHRM → INNOVACAP | 0.20 ** | 26.161 | | Supported |
| H2—INNOVACAP → COMPAD | 0.70 *** | 23.949 | | Supported |
| H3—MGHRM → COMPAD | 0.13 ** | 13.575 | | Supported |
| COMPAD | | | 0.723 | |

Model fit: ($\chi^2$ (50, N = 558) = 88,721 $p < 0.001$, normed $\chi^2$ = 1774, RMSEA = 0.037, RMR = 0.022, SRMR = 0.0376, GFI = 0.946, CFI = 0.966, NFI = 0.926, RFI = 0.902, TLI = 0.955, PCFI = 0.732 and PNFI = 0.701), *** $p < 0.001$; ** $p < 0.01$.

**Table 6.** The results of mediation in relation to females (developed by authors).

| User-Defined Estimands: (Group Number 1—Default Model) | | | | | Mediation |
|---|---|---|---|---|---|
| Parameter | Estimate | Lower | Upper | $p$ | Perfect Mediation |
| H4—FGHRM → INNOVACAP → COMPAD | 0.113 | 0.020 | 0.272 | 0.056 | 0.056 > 0.05 |

**Table 7.** The results of mediation in relation to males (developed by authors).

| User-Defined Estimands: (Group Number 1—Default Model) | | | | | Mediation |
|---|---|---|---|---|---|
| Parameter | Estimate | Lower | Upper | $p$ | Partial Mediation |
| H4—MGHRM → INNOVACAP → COMPAD | 0.103 | 0.044 | 0.175 | 0.003 | 0.003 < 0.05 |

However, concerning the male entrepreneurs' model, "The user-defined estimands", it is shown (Table 7) that a positive and significant link between green human resource management and competitive advantage even after the introduction of the mediator variable (β = +0.103, $p$ = 0.003 < 0.05). Therefore, we can conclude that the mediation by the innovative capacity proposed by male entrepreneurs is, therefore, partial between green human resource management and competitive advantage. Moreover, the Sobel test gives us a z-value equal to 4.226 > 1.96 with a $p$-value of zero, i.e., less than 0.01. The research's final structural models are shown in Figures 1 and 2 for females and males, respectively.

## 5. Focus Group, Discussion and Implications

In order to explain our results, we approached the owner–managers of some companies to carry out semi-structured interviews with 10 interviewees with two different groups. The first group has five women and the second group has five men. All of them were owner–managers of their companies. We have coded the names of interviewees and their companies to protect their privacy. The results obtained from the content analysis of the interviews [65] prompted us to ask more questions via focus groups to carry out an empirical immersion. The answers to our questions are seen as contributions to the theoretical and managerial levels. Moreover, it goes without saying that we were interested, as recommended by Schütz [66], in the common language, which hides a treasure of social quintessence, including hidden questions, which is our responsibility to disclose. The responses from the interviews allowed us to understand why the model of the women entrepreneurs is perfectly mediated, in contrast to the model of their male counterparts. Based on a redundancy of certain words or groups of words evoked by the female and male interviewees during the conduct of the interviews, we highlighted a narrative content grid. The thematic analysis of the interviewees' verbatim allowed us to highlight two categories to which two lexical fields correspond, which we will develop. The first category refers to a "purely cultural problem", and the second category refers to "the leadership style that differs from one gender to another". These categories are organized according to their order of importance in the respondents' comments. We discuss the main themes of the interviews in the next two sections (Sections 5.1 and 5.2).

### 5.1. From Deprivation Comes Motivation

Saudi women are consistently motivated. However, women have a more difficult start to their careers. In this respect, the interviewees unanimously stated that they did not understand why women were almost absent from the organization's management bodies. Where issues with major repercussions are discussed, such as finance, strategic decisions, crucial investments, etc., Ms. N, CEO of six outlets in Alkhobar, says: "*I expected so much from my former superiors, that I would get even moral recognition or real support from them, but in vain. Now that I am at the top of the pyramid, I have vowed to pay more attention to all these little details with regard to my male and female subordinates, regardless of their nationalities*".

In this respect, the role of prejudices and organizational or even social stereotypes can be estimated as having a negative influence on cognitive filters and behavior. Bourdieu [67], in his book "La domination masculine" (male domination), showed how the so-called natural differences between the two genders are socially and historically constructed. However, this view contrasts the belief that gender differences are attributable to biological differences, or a mix of both social construction and biological attributes. This feeling of devaluation has been an unwavering motivation for women entrepreneurs to be the best at what they do. These women are inspiring in many ways. Whether it is through their product or service, their dedication or their leadership with their employees, they are examples of successful businesses.

Moreover, some women entrepreneurs organize whole days of training for their employees on subjects related to respect for nature. Others showed us their efforts to invest in the environment by allowing their customers to use environmentally friendly packaging. "*It's a way of building loyalty among a category of our clientele that is concerned about nature and respect for the environment*", says Miss F, owner of two restaurants in Riyadh. Indeed, by adopting eco-responsible packaging, these companies become eco-responsible in the eyes of their customers, which is their competitive advantage par excellence, and which is materialized not only by more sales and more profits, but also by the possibility of taking advantage of their influence to modify the behavior of potential consumers, or even prospects. There was consensus among the interviews that women are more likely to pay attention to environmental concerns and green issues than their male counterparts are.

### 5.2. A Year Is 365 Ideas and More

All the women interviewed start from the assumption that the way an employee is viewed can dictate his or her behavior. That is, they may behave passively, if they have been considered passive in the first place. This will certainly have a negative impact on the smooth running of the project and its ability to develop further. We note here that where women entrepreneurs are more willing to implement green human resource practices, male entrepreneurs are more bureaucratic or even Taylorian in nature. This issue, however, cannot be simply generalized, but needs further investigation and could be just the opinion of our interviewees. It turns out that female entrepreneurs are more empathetic to their subordinates than their male counterparts are to theirs.

Believing strongly in the principle that people should be at the center to ensure more social innovation, these firstly offer a friendly but challenging environment. Hence, the breathtaking flow of innovative ideas. Ms. R, owner of four Teashops, says, "*regardless of nationality and rank, everyone has the right to propose ideas and implement them, as it is said that from diversity comes progress*". Indeed, from the indirect questions we asked, it is clear that information is more fluid and bottom-up in women-led companies than in male-led companies. Interestingly, the results of interviews showed that unlike male entrepreneurs, female entrepreneurs praise work well done; encourage creativity and initiative; disseminate information to all employees; act impartially; know how to look for the right incentives to get everyone on board; are more sensitive to the present and future needs of their employees; inspire enthusiasm, which in turn inspires effort; and prove their confidence by delegating responsibility to subordinate employees. This could be because

recent empowerment, as a part of the Saudi vision 2030, and hence they would like to express themselves in the Saudi society.

In the end, female leaders were found to pay more attention to their followers than their males counterparts. This concurs with the writings of Ishak et al. [68], who argue that effective HRM should ensure a balance between organizational systems that, on the one hand, are sufficiently open and flexible to allow creativity, but on the other hand have sufficient formality and discipline for creativity to produce tangible results. This balance achieved by female entrepreneurs creates a space for unrestricted creativity, thus ensuring continuous innovation that can support a proven competitive advantage. In contrast, the bureaucracy and formal communication applied by most of our sample of male entrepreneurs inhibits the spontaneity and freedom of expression necessary for innovative responses to rapid environmental change. This insightfully could justify the perfect mediation of innovation capacities found in the results of quantitative empirical research specific to female entrepreneurs, in contrast to male entrepreneurs, whose mediation of innovation capacities is only partial in the relationship between green human resource management and maintaining competitive advantage.

The current research study support the literature discussed earlier on the relationship between GHRM, innovation capacity and competitive advantages. More specifically, our research supported previous research [25–29] that GHRM has a positive and significant effect on innovation capacities in the organizations. Our research also confirmed a positive significant influence of GHRM on the competitive advantage of food companies, which support the work of previous researchers [35–41]. Moreover, our research confirmed a positive direct influence of innovation on competitive advantage, which also supports research studies [43–45]. Furthermore, our research showed a perfect mediation role for female entrepreneurs of their innovation in the relationship between GHRM and competitive advantage. On the other side, the research confirmed a partial mediation role for male entrepreneurs of their innovation in the relationship between GHRM and competitive advantage. As discussed above, in the results of group interviews, female entrepreneurs were found to give more freedom to their subordinates and encourage their creativity than their male counterparts, which could justify this perfect mediation in the relationship.

Our research has several implications for scholars and policy makers. First, our research extends the literature on RBV theory by confirming that GHRM has direct positive influence on both innovation and competitive advantages for both males and female entrepreneurs. The literature [22] confirmed that investing in HR could allow organizations to achieve competitive advantage. Our research extended this concept and showed that the green practices by entrepreneur "leaders" with their HR investment encourage innovation and allow organizations to achieve competitive advantages. Second, the research contributes to the literature and bridges a gap in knowledge in relation to the differences between male and female entrepreneurs in their GHRM and its impact on innovation and competitive advantages. Our research confirmed a partial mediation role for male entrepreneurs and perfect mediation for female entrepreneurs in relation to the mediating role of innovation in the relationship between GHRM and competitive advantage. This highlights the heterogeneity between male and female entrepreneurs, which has to be recognized by scholars when studying entrepreneur orientation based on gender. The results can be useful to policymakers in Saudi Arabia that women empowerment can be fruitful. More empowerment of women entrepreneurship may add economic, social and environmental value to the Saudi society. In line with the accelerated recent progress of women empowerment associated with vision 2030 in Saudi Arabia, policy makers should continue empowering Saudi women entrepreneurship.

## 6. Conclusions

This research is among the first attempts to compare the green human resource management practices of female entrepreneurs with those of their male counterparts and the

impacts of this on the innovation capacities of each gender and on the competitive advantages in the food industry in Saudi Arabia. The results showed positive significant influences on GHRM on both innovation capacities and competitive advantages. Additionally, our research showed a perfect mediation role for female entrepreneurs of their innovation in the relationship between GHRM and competitive advantage. However, there was a partial mediation role for male entrepreneurs of their innovation in the relationship between GHRM and competitive advantage. The research extends the literature on the influences of GHRM, especially when it comes to gender role. Understanding this role and differences between male and female enable scholars recognizing the heterogeneity of entrepreneurs. It will also allow policymakers to have better regulations, whether environmental, social or economic, for their entrepreneurs to achieve national agenda.

## 7. Limitations and Future Research Opportunities

It is true that this study has enabled us to meet the objectives we set ourselves. Nevertheless, as in all research work, several limitations need to be highlighted. Firstly, we start with the limitations of studying a specific population. Indeed, the results we have reached depend solely on SMC. Thus, the findings we have collected do not allow us to generalize the results obtained, especially when we talk about other contexts or countries different from Saudi Arabia. Secondly, there are limitations related to the declarative nature of the data collection methods. Since the qualitative mini research was not based on participant observation in the different SMC we visited, but on a set of representations of the respondents' reality. Although these limitations do not challenge the results, it opens the door for future research opportunities. For example, the variables of corporate social responsibility and culture could be included in the further research proposed models. Certainly, the concept of ecology can be integrated following some good practices.

**Author Contributions:** Conceptualization, H.G., A.E.E.S., N.A. and A.A.; methodology, H.G. and A.E.E.S.; software, H.G; validation, H.G. and A.E.E.S.; formal analysis, H.G., N.A. and A.E.E.S.; investigation, H.G. and A.E.E.S.; resources, H.G., A.E.E.S., N.A. and A.A.; data curation, H.G.; writing—original draft preparation H.G., A.E.E.S., N.A. and A.A.; writing—review and editing, H.G., and A.E.E.S.; visualization, H.G. and A.E.E.S.; supervision, A.E.E.S.; project administration, H.G., A.E.E.S., N.A. and A.A.; funding acquisition, H.G. All authors have read and agreed to the published version of the manuscript.

**Funding:** This work was supported by The Saudi Investment Bank Scholarly Chair for Investment Awareness Studies, the Deanship of Scientific Research, Vice Presidency for Graduate Studies and Scientific Research, King Faisal University, Saudi Arabia (Grant No. CHAIR80).

**Institutional Review Board Statement:** The study was conducted according to the guidelines of the Declaration of Helsinki and approved by the deanship of scientific research ethical committee, King Faisal University (project number: CHAIR80, date of approval: 1 January 2022).

**Informed Consent Statement:** Informed consent was obtained from all subjects involved in the study.

**Data Availability Statement:** Data are available upon request from researchers who meet the eligibility criteria. Kindly contact the first author privately through the e-mail.

**Conflicts of Interest:** The authors declare no conflict of interest.

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
