# Peer review of "The Role of Innovation Capacities in the Relationship between Green Human Resource Management and Competitive Advantage in the Saudi Food Industry: Does Gender of Entrepreneurs Really Matter?"

_agriculture, doi:10.3390/agriculture12060857_

Round 1
Reviewer 1 Report
The paper has following several shortcomings
-There is no number of section 3.
-There is no Conclusion section of the paper.
-There is no Policy implications
-Results are not presented well. Also the results of this paper are not compared with the previouly published.
Author Response
Dear Reviewer,
Thank you for giving us the opportunity to resubmit a revised draft of our manuscript titled “The Role of Innovation Capacities in the Relationship between Green Human Resource Management and Competitive Advantage in the Saudi Food Industry: Does Gender of Entrepreneurs really Matter?” for publication consideration in a special issue in the elite journal “Agriculture”.
We appreciate the time and effort that you have dedicated to providing your valuable feedback on our manuscript. We are grateful for your insightful comments on our paper. We have been able to incorporate changes to reflect the suggestions provided in our revised manuscript. We have used track changes to easily check changes.
We have attached a point-by-point response to your comments and concerns in the attached file.
Best regards
the authors

Reviewer 2 Report
Thank you for the opportunity to read this interesting and current article aimed the Role of Innovation Capacities in the Relationship between Green Human Resource Management and Competitive Advantage in the Saudi Food Industry: Does Gender of Entrepreneurs really Matter?. The topic of the paper is relevant and original in respect to the field of application. Results are relevant from both a scientific, managerial and political point of view. The manuscript is also well organized and the results are well discussed. However, the quality of the paper could be improved to give adequate value to the research. Specific comments are reported below.
The section “Introduction” must be improved. A greater effort is required to the authors in order to significantly improve this section in which the authors must emphasize non only the research background but in particular I suggest to includes in this section interesting sources such as:
Fiore, M., Galati, A., Gołębiewski, J., & Drejerska, N. (2020). Stakeholders' involvement in establishing sustainable business models: The case of Polish dairy cooperatives. British Food Journal.
Chouaibi, S., Rossi, M., Siggia, D., & Chouaibi, J. (2021). Exploring the moderating role of social and ethical practices in the relationship between environmental disclosure and financial performance: evidence from ESG companies. Sustainability, 14(1), 209.
Section “Methodology” is well organized and presented.
Results are clearly presented. A greater effort is required to the authors in order to improve this section in which the authors should compare their results with previous empirical findings (if there are).
Finally, in the section “Conclusion” in which the authors summarize the main findings, I suggest to conclude the work also with Implications, limitations and future research.
Author Response
Dear Reviewer,
Thank you for giving us the opportunity to resubmit a revised draft of our manuscript titled “The Role of Innovation Capacities in the Relationship between Green Human Resource Management and Competitive Advantage in the Saudi Food Industry: Does Gender of Entrepreneurs really Matter?” for publication consideration in a special issue in the elite journal “Agriculture”.
We appreciate the time and effort that you have dedicated to providing your valuable feedback on our manuscript. We are grateful for your insightful comments on our paper. We have been able to incorporate changes to reflect the suggestions provided in our revised manuscript. We have used track changes to easily check changes.
We have attached a point-by-point response to your comments and concerns in the attached file.
Best regards
The authors

Reviewer 3 Report
The manuscript addresses the important and timely issue of green human resource management and innovation. This means that it may be of interest to Agriculture readers. The purpose of the research was to examine the mediating role of innovation capacities in the relationship between green human resource management and competitive advantage in the Saudi food industry. The authors set out 4 research hypotheses.
The article is interesting and prepared to a high standard. However, it requires minor improvements:
1) Line 77 - the chapter number is missing.
2) Line 84 and 104 - page number in bibliographic reference is unnecessary.
3) More discussion with the results of other authors is worthwhile.
4) In the abstract the authors wrote "The results of our research has 30 numerous implications and both scholars and policymakers, especially in relation to the Saudi food industry". However, direct recommendations from the research are missing.
Author Response

(The authors gave the same response as above.)

Round 2
Reviewer 1 Report
The paper can be accepted in current form.
Author Response
Dear Reviewer,
Thank you for positive feedback a revised draft of our manuscript titled “The Role of Innovation Capacities in the Relationship between Green Human Resource Management and Competitive Advantage in the Saudi Food Industry: Does Gender of Entrepreneurs really Matter?” for publication consideration in your special issue in the elite journal “Agriculture”.
We appreciate the time and effort that you have dedicated to providing your valuable feedback on our manuscript. We are grateful for your insightful comments on our paper. We have been able to respond to each comment in this version and incorporate changes to reflect the suggestions provided in our revised manuscript. We have used track changes to easily check changes.
We also have considered all of their comments in our revised manuscript.
Hope it meets your expectations.
Thanks for your consideration and looking forward to hearing from you.
Abu Elnasr Sobaih